# High Glucose Concentrations Negatively Regulate the IGF1R/Src/ERK Axis through the MicroRNA-9 in Colorectal Cancer

**DOI:** 10.3390/cells8040326

**Published:** 2019-04-08

**Authors:** Ya-Chun Chen, Ming-Che Ou, Chia-Wei Fang, Tsung-Hsien Lee, Shu-Ling Tzeng

**Affiliations:** 1Institute of Medicine, Chung Shan Medical University, Taichung 402, Taiwan; big383838@hotmail.com; 2Department of Hematology and Oncology, Tungs’ Taichung MetroHarbor Hospital, Taichung 435, Taiwan; mingche.ou@gmail.com; 3Division of Colon and Rectal Surgery, Department of Surgery, Taichung Tzu Chi Hospital, Buddhist Tzu Chi Medical Foundation, Taichung 427, Taiwan; forgive603@yahoo.com.tw; 4Department of Obstetrics and Gynecology, Chung Shan Medical University Hospital, Taichung 402, Taiwan

**Keywords:** high concentration of glucose, microRNA-9, colorectal cancer, insulin-like growth factor-1 receptor, Src, extracellular signal-regulated kinase (ERK1/2), proliferation, metastasis, epithelial to mesenchymal transition

## Abstract

Studies have revealed that people with hyperglycemia have a high risk of colorectal cancer (CRC). Hyperglycemia may be responsible for supplying energy to CRC cells. However, the potential molecular mechanism for this association remains unclear. Furthermore, microRNA-9 (miR-9) has a tumor-suppressive function in CRC. Aberrant reduced expression of miR-9 is involved in the development and progression of malignancy caused by a high glucose (HG) concentration. In this study, we used an HG concentration to activate miR-9 downregulation in CRC cells. Our results indicated that miR-9 decreased the insulin-like growth factor-1 receptor (IGF1R)/Src signaling pathway and downstream cyclin B1 and N-cadherin but upregulated E-cadherin. The HG concentration not only promoted cell proliferation, increased the G1 population, and modulated epithelial-to-mesenchymal transition (EMT) protein expression and morphology but also promoted the cell migration and invasion ability of SW480 (low metastatic potential) and SW620 (high metastatic potential) cells. In addition, low glucose concentrations could reverse the effect of the HG concentration in SW480 and SW620 cells. In conclusion, our results provide new evidence for multiple signaling pathways being regulated through hyperglycemia in CRC. We propose that blood sugar control may serve as a potential strategy for the clinical management of CRC.

## 1. Introduction

Colorectal cancer (CRC) is one of the major malignancies worldwide and the third leading cause of cancer death in the United States [1,2]. Epidemiological data demonstrated significant increases of various cancers in people with obesity and diabetes [3]. In Taiwan, diabetes is associated with an increased risk of CRC and worse disease progression [4]. Hyperglycemia is often wrongly implicated as the sole source of cancer nutrition in patients with diabetes; cancer cells can thrive using other energy sources promoted by genetic mutations and aberrant intracellular signaling. Hyperglycemia is an independent risk factor for CRC [5,6]. Because of the prevalence of hyperglycemia-related conditions in patients with cancer, the relationship between hyperglycemia and cancer should receive adequate attention. Notably, for both normal cells and cancer cells, a high glucose (HG) concentration may contribute to enhancing cell proliferation ability and cell cycle progression. Only cancer cells may be modulated to alter cell metastasis, perineural invasion, and chemotherapy resistance and intolerance through HG concentrations [5,6,7,8,9]. The mechanistic role of glucose in CRC cells is poorly understood.

MicroRNAs (miRNAs) are small noncoding RNAs that aberrantly regulate the epigenome in all tumor types [10,11]. Several mechanisms exist that show that microRNAs regulate the epigenome post-transcriptionally in response to gene expression [11]. MiRNAs regulate varioustarget genes involved in multiple pathways and processes, such as development, apoptosis, proliferation, differentiation, transformation, and cellular senescence [12,13,14]. The involvement of miRNAs in cancer pathogenesis is now well established, and increasing evidence exists that they can behave either as onco-miRNAs or tumor-suppressive miRNAs [15,16,17]. Research suggested that microRNA-9 (miR-9) expression levels were remarkably reduced in CRC tissues compared with those in matched normal tissues [18]. In addition, Bandres et al. identified that DNA hypermethylation and histone modifications contribute to the transcriptional downregulation of miR-9 and may participate in human colorectal tumorigenesis [19,20].

HG concentrations were shown to promote CRC progression [20]. Approximately 90% of all cancer deaths are linked to tumor metastasis. Hyperglycemia, through ligands binding to insulin receptors, may increase circulating insulin-like growth factor-1 (IGF-1) levels [21,22]. IGF-1 was implicated as a key factor in the mechanisms involved in carcinogenesis [23]. In parallel, IGF-1 receptor (IGF1R) is an autophosphorylated receptor that binds to the Src homology domain and insulin receptor substrate (IRS). IGF1R activates the IRS protein SHC, which then stimulates Raf before triggering a kinase cascade, eventually resulting in the activation of mitogen-activated protein kinases(MAPKs) and extracellular signal-regulated kinases 1 and 2 (ERK1 and ERK2, respectively) through the GTPase Ras [24]. Src overexpression has been shown to increase cell adhesion, invasion, and migration in CRC cells. In addition, ERK1/2 may influence transcriptional factors, leading to increased cell cycle activity and promoting cancer progression [25,26]. In this study, we focused on whether a specific HG concentration can influence cancer cell proliferation and metastasis in CRC through the miR-9-IGF1R or Src pathway. We attempted to clarify the modulating effect of these signaling pathways. Our findings may provide new insights into the molecular mechanisms through which HG-concentration environments influence CRC as well as reveal a novel therapeutic strategy for patients with CRC who simultaneously have hyperglycemia.

## 2. Materials and Methods

### 2.1. Cell Culture

To determine how HG concentrations could influence epithelial-to-mesenchymal transition (EMT) activities and cause changes in signal cascade activities involved in the migration of cancer, the human colon cancer cell lines SW480 (no.CCL-228; ATCC^®^) and SW620 (no.CCL-227; ATCC^®^) were maintained in Dulbecco’s modified Eagle’s medium (DMEM; Gibco^®^ cat.11995-040 and Gibco^®^ cat.11885-076) supplemented with 10% inactive fetal bovine serum (FBS; cat.SH30071.03; HyClone^®^) and 100 U/mL of penicillin/treptomycin. Cellular suspensions were obtained through incubation with 0.5 mL of 0.5% trypsin-EDTA (cat.15400-054; Gibco^®^) for 2–5 min and cultured in an incubator in a humidified atmosphere of 5% CO_2_ at 3 °C.

### 2.2. Drug

The IGF1R inhibitor OSI-906 was purchased from Selleckchem^®^ (#S1091), and the Src inhibitor PP1 was obtained from Calbiochem^®^ (#567809). OSI-906 or PP1 was dissolved in 100 mM or 4 mM DMSO and stored at −80 °C for in vitro studies. The cells were grown overnight and treated with 1.0 and 2.5 μM OSI-906 or 2.0 and 4.0 μM PP1, and then data were collected using a trypan blue assay, Western blotting, and a migration and invasion assay.

### 2.3. Transient Transfection

Approximately 3 × 10^5^ SW480 or SW620 cells were seeded onto 3.5-cm dishes for 24 h before transfection. Lipofectamine 2000 was used according to the manufacturer’s protocol (Invitrogen, Thermo Fisher Scientific, Waltham, MA, USA). After one night, cells were transferred to fresh incomplete medium for 20 min before transfection. DNA mixture and Lipofectamine 2000 were prepared, and then DNA particles were added to the cells. After transfection for 2 h (SW480 cells) or 5 h (SW620 cells), the complete medium was refreshed before being incubated at 37 °C. After the cells had been cultured for 48 h, they were detected using qRT-PCR.

### 2.4. Western Blotting

A polyvinylidene difluoride (PVDF) membrane (Millipore) and Whatman 3MM paper were cut to sizes equal to sodium dodecyl sulfate–polyacrylamide (SDS-PAGE) gels. The PVDF membrane was immersed in methanol for 1 min, in ddH_2_O for 2 min, and finally in transfer buffer (25 mM Tris base, 192 mM Glycine, 15% Methanol) for 5 min. After SDS-PAGE was executed, the SDS-PAGE gel was immersed in transfer buffer for 10 min. After the transfer, the membrane was first incubated in NET blotting solution (0.15 M NaCl, 5 mM EDTA-2Na, 50 mM Tris, 0.25% gelatin, and 20% Tween 20) at 37 °C with gentle shaking for 30 min to block nonspecific binding; it was then incubated with primary antibodies in blotting solution at 4 °C overnight. The membrane was washed in 1X tris-buffered saline with tween-20 (TBST) blotting solution for 5 min three times, after which it was incubated with horseradish-peroxidase-conjugated secondary antibodies for 50 min at room temperature. The mentioned washing process was repeated, after which bound antibodies were detected using an enhanced chemiluminescence system according to the manufacturer’s instructions (Millipore Corporation, Billerica, MA, USA). The primary antibodies anti-N-cadherin Ab (J94353; St John’s), anti-E-cadherin Ab (J92819; St John’s), anti-Src Ab (J40571; St John’s), anti-IGF1R Ab (J31780; St John’s), anti-IGF1Rβ (pY11135/1136) Ab (#3024; Cell Signaling Technology), anti-Vimentin Ab (NBP1-92687SS; Novus), anti-Src (pY418) Ab (44660G; Invitrogen), anti-cyclin D1 (A-12) Ab (sc-8396; Santa Cruz), anti-P16 Ab, anti-p53 (FL-393) Ab (sc-6243; Santa Cruz), and anti-CDC42 (B-8) Ab (sc-8403; Santa Cruz) were imaged using a biomolecular imager (LAS-4000; GE Healthcare).

### 2.5. RNA Extraction and qRT-PCR Using Vector Constructs and Transfection

The miRNA expression vectors for pre-miR-9 were obtained from Applied Biosystems. The expression of the human pre-miR-9 sequence was amplified from human genomic DNA using the specific primer UCUUUGUUAUCUAGCUGUAUGA (#4427975; Applied Biosystems) through qRT-PCR. Subsequently, 1 × 10^5^ SW480 or SW620 cells were seeded onto 3.5-cm dishes. After 24 h of culture, the cells were transfected with miR-9 mimics or NC using the reagent Lipofectamine 2000 according to the manufacturer’s instructions (Invitrogen, Thermo Fisher Scientific, Waltham, MA, USA). After 48 h of culture, the cells were detected using qRT-PCR. Total RNA was extracted using Trizol (Invitrogen), and qRT-PCR analyses of miR-9 and RNU6B (housekeeping control) were conducted using TaqMan^®^ microRNA assays (Applied Biosystems). Appropriate dilutions of each cDNA for subsequent PCR amplification were determined through TaqMan^®^ Universal PCR Master Mix and TaqMan^®^ Small RNA assay. The relative quantification of expression was performed using the 2^−∆∆^ Ct method in the StepOnePlus software (Thermo Fisher Scientific, Waltham, MA, USA) package. A t test was used to determine statistical significance. All reactions were initially denatured at 95 °C for 10 min followed by 45 cycles at 95 °C for 15 s and 72 °C for 60 s on a StepOnePlus system (Applied Biosystems).

### 2.6. Cell Cycle Analysis

A total of 1 × 10^5^ cells were plated onto 3.5-cm dishes at a glucose concentration of 5.5 mΜ (NG) or 25 mΜ (HG) for 24 h. After starvation for 24 h, the cells were transferred to complete medium for another 24 h. Subsequently, they were suspended, washed with 1× phosphate buffered saline, and collected by centrifugation. The pellets were mixed with 75% ethanol for 1 min at −20 °C. The cells were then centrifuged and resuspended in 500 μL of 10 mg/mL RNase A for 10 min at 37 °C. Next, they were treated with 1 mg/mL of PI staining solution at 37 °C in darkness before being analyzed using a flow cytometer (FACSCalibur^TM^, BD Biosciences) (San Jose, CA, USA,).

### 2.7. Cell Migration and Invasion Assays

Cell migration and invasion assays were performed using Millicell^®^ inserts with a pore size of 8 μm. First, SW480 or SW620 cells were seeded onto a 24-well plate at a density of 3 × 10^5^ cells per well. The plate was incubated to establish confluent monolayers. The cells were seeded into serum-starving medium (0.1% FBS DMEM) in the upper chambers of an insert. Subsequently, cells were added to the lower chamber containing 10% FBS DMEM medium (BD Biosciences) for 96 h or 120 h. For invasion, inserts were coated in CULTREX^®^ Basement Membrane Extract at 37 °C for 30 min according to manufacturer’s protocol. Both assays of cells in the upper chamber were removed, and the attached cells that had migrated or invaded into the lower section were fixed and stained with 0.1% crystal violet. Images of stained cells were captured at OD_595_ using a microplate reader (SpectraMax M5; Molecular Devices).

### 2.8. Wound Healing

Next, 3 × 10^5^ SW480 or SW620 cells were seeded and wounded onto a 3.5-cm dish using cell culture inserts (ibidi^®^). Images of the cells were captured under an inverted microscope at 0, 24, 48, 72, and 96 h after wounding. Images of invaded cells were captured randomly from selected fields using an inverted microscope.

### 2.9. Colorectal Tissue Preparation

Tumor and adjacent tissue samples were collected from 13 patients with CRC who had undergone curative surgical resection at Taichung Tzu Chi Hospital and received a pathological diagnosis of colon adenocarcinoma. Tissue samples were frozen immediately in liquid nitrogen and stored at −80 °C before use. All patients provided informed consent, and the study was approved by the Scientific Ethics Committee of Taichung Tzu Chi Hospital (REC102-21). Colorectal tissue from each specimen was artificially homogenized for 20 s three or four times. Samples were then lysed in RIPA buffer supplemented with 100 mM phenylmethylsulfonyl fluorid (PMSF) in EtOH, 1 mg/mL of leupeptin in H_2_0_2_, and 1 mg/mL of aprotinin in H_2_0_2_ or extracted using Trizol for 5 min of reverse transcription (Invitrogen) and then centrifuged. Furthermore, total protein was measured using the Bradford protein assay (Bio-Rad, Hercules, CA, USA) and stored at −20 °C.

### 2.10. Statistical Analysis

Correlations between the two groups were analyzed through independent study. All data were analyzed for significant differences by using Student’s *t*-tests. The results are presented as the mean ± standard deviation (error bars). All experiments were performed at least in duplicate, and *p* values of <0.05 were considered statistically significant.

## 3. Results

### 3.1. d-glucose Promoted Cell Proliferation and Increased Cell-Cycle-Regulated Protein Expression in CRC Cells

Glucose is an essential source of energy and nutrients for the growth and survival of normal cells and cancer cells. In a medium, a glucose concentration of 5.5 mM corresponds to normal physiological levels in human blood (100 mg/dL), whereas a concentration of 25 mM (approximately 450 mg/dL) is equivalent to severe hyperglycemia [27]. To test the effect of glucose on the growth of CRC cells, we cultured SW480 (low metastatic potential) and SW620 (high metastatic potential) cells in medium with three different glucose concentrations for between 0 and 120 h: Physiologically normal glucose (NG) concentration (5.5 mM d-glucose), HG concentration (25 mM), and normal concentration plus l-glucose (NG + l-glucose; 5.5 mM d-glucose + 19.5 mM l-glucose). The results showed that cell proliferation increased by 1.59-fold (*p* < 0.005) and 2.54-fold (*p* < 0.005) at 120 h in SW480 and SW620 cells cultured using the HG concentration, respectively, compared with those cultured using the NG and NG + l-glucose (Figure 1A,B). These results indicate that d-glucose but not l-glucose promoted cell proliferation. Moreover, the results suggest that d-glucose might induce CRC cell growth. To determine whether the HG concentration increased cell proliferation compared with the NG, 1 × 10^5^ cells were seeded onto a 3.5-mm dish for 24 h of serum starvation. We measured DNA synthesis through propidium iodide incorporation at 24 h using a flow cytometer (FACSCalibur^TM^, BD Biosciences). The HG concentration increased the G1 population from 49.2% to 61.0% in SW480 cells (*p* < 0.05) and from 55.0% to 62.1% in SW620 cells (*p* < 0.005) (Figure 1C,D). Therefore, HG concentrations may enhance cell proliferation. Our observations showed that the cell cycle regulatory proteins CDC42, cyclin B1, cyclin D1, and p16 were significantly increased but that p53 was unchanged by Western blotting (Figure 1E). This indicates that the HG concentration increased cell proliferation through enhanced cell cycle progression in both early-stage SW480 and advanced-stage SW620 cells in CRC.

### 3.2. HG Concentration Induced Epithelial-to-Mesenchymal Transition Protein Expression and Enhanced Migration Activity in CRC Cells

To determine how HG concentrations could influence epithelial-to-mesenchymal transition (EMT) activities and cause changes in signal cascade activities involved in the migration of cancer cells, we further tested the possibility that HG concentrations might be involved in controlling EMT in CRC cells. We cultured SW480 and SW620 cells in medium with different concentrations of glucose (NG, HG, and NG + l-glucose). We demonstrated that high concentrations of d-glucose not only promoted cell proliferation but also induced a morphological change from epithelial to mesenchymal type (Figure 2A). According to the results of a Western blot assay, the HG concentration caused the downregulation of E-cadherin and upregulation of N-cadherin, β-catenin, and vimentin (Figure 2B); however, c-myc was unchanged. Because migration assays are excellent indicators of long-term tumor cell metastasis, we subsequently assessed whether the HG concentration had an effect on the migration ability of SW480 and SW620 cells. Using a wound healing assay, we observed that the HG concentration promoted SW480 and SW620 cell motility compared with the NG and NG + l-glucose groups after 48 and 72 h of culture (Figure 2C,D). Images captured using an inverted microscope under 100× magnification revealed invaded cells (black) on the Matrigel surface. As expected, the results showed that the HG concentration significantly increased the migration of SW480 and SW620 cells by 1.85-fold (*p* < 0.05) and 2.05-fold (*p* < 0.005) at 96 h, as determined using a Transwell assay (Figure 2E). These results are in agreement with those of our previous studies demonstrating that HG concentrations induced changes from epithelial to mesenchymal form (Figure 2A). We further observed that the HG concentration significantly upregulated p-IGF1R (pY11135/1136) protein levels in SW480 and SW620 cells by 1.68-fold (*p* < 0.005) and 1.52-fold (*p* < 0.005), respectively, as determined using Western blotting (Figure 2F). In addition, the HG concentration promoted downstream signaling proteins, including p-Src (pY418) and p-ERK, in CRC cells (Figure 2F).

### 3.3. HG Concentration Regulated IGF1R and Src and Promoted Downstream Signaling Pathways in CRC Cells

Based on the results presented in Figure 1; Figure 2, multiple regulatory signaling pathways might be involved in the mechanisms through which HG concentrations affect CRC. Therefore, we investigated the signaling mechanisms through which the HG concentration stimulated cell proliferation and migration through IGF1R and Src in human CRC cells. Our results showed that OSI-906 (IGF1R inhibitor) decreased the rate of cell proliferation in a dose-dependent manner at 1.0 and 2.5 μM. Furthermore, we found that OSI-906 inhibited HG-concentration-induced IGF1R-activity and cell proliferation in SW480 cells at doses of 1.0 μM (*p* < 0.05) and 2.5 μM (*p* < 0.005), and in SW620 cells at doses of 1.0 μM (*p* < 0.05) and 2.5 μM (*p* < 0.05) (Figure 3A,B). PP1 (Src inhibitor) inhibited the effect of the HG concentration in a dose-dependent manner at 2.0 and 4.0 μM. According to the results of a trypan blue assay, we selected a concentration of 2.0 μM and a time point of 48 has adequate intervention parameters for subsequent experiments. Our results showed that PP1 treatment caused decreased cell growth in SW480 cells (*p* < 0.005) and in SW620 cells (*p* < 0.05) (Figure 3C,D). Therefore, we further examined whether IGF1 and Src activity affected HG-concentration-enhanced migration and invasion ability as well as induced downstream protein levels in CRC cells. Statistical analysis revealed that OSI-906 (2.5 μM) and PP1 (2.0 μM) remarkably decreased HG-concentration-induced cell migration ability compared with the control group (dimethyl sulfoxide, DMSO) in SW480 and SW620 cells (Figure 3E–H). Notably, OSI-906 decreased N-cadherin and reduced cyclin B1, but only cyclin B1 and E-cadherin were unchanged in SW620 cells (Figure 3I). Similarly, PP1 reduced cyclin B1 (Figure 3J) compared with the control group (DMSO) cultured in HG-concentration medium. Thus, high levels of IGF1R are associated with increased incidence of cancer progression, whereas lower levels of IGF1R are associated with decreased incidence of cancer progression in CRC cell lines. In our CRC cell lines, as expected, IGF1R inhibition appeared to primarily act through the Src (pY418) ERK phosphorylation signaling pathway. The results showed significant HG-concentration-induced IGF1R (pY11135/1136), p-Src (pY418), and p-ERK activation as well as EMT development (Figure 2A–F). These data clearly demonstrate that the HG concentration promoted cell proliferation, migration, and invasion ability through the IGF1R/Src/ERK pathway.

### 3.4. Expression and Regulation of miR-9 in CRC Cell Lines by HG Concentration

Next, we investigated whether the HG concentration could induce miR-9 to affect the IGF1R/Src pathway and functionality. TaqMan quantitative reverse transcription polymerase chain reaction (qRT-PCR) analysis was performed on miR-9 in SW480 and SW620 CRC cell lines that were cultured in various concentrations of glucose (NG and HG). The HG concentration significantly downregulated miR-9 levels in SW480 and SW620 CRC cells by 0.59-fold (*p* < 0.005) and 0.31 fold (*p* < 0.05), respectively (Figure 4A,B). In our previous study, we investigated how hyperglycemia can potentially increase the risk of CRC in premalignant lesions and enhance cancer progression in clinical patients [6]. In the present study, we prepared transfected SW480 and SW620 cells with miR-9 mimics at different doses (15 and 30 nM) or an miR mimic negative control (NC) for 48 h in the HG-concentration medium (Figure 4C,D). Our results demonstrated that the miR-9-transfected cells expressed lower levels of the proteins p-IGF1R (pY11135/1136), cyclin B1, and N-cadherin; however, E-cadherin was more upregulated in the NC-transfected SW480 and SW620 cells. Thus, miR-9 is a tumor-suppressive microRNA that may inhibit the IGF1R pathway to regulate the targeting of cyclin B1 and N-cadherin, and increase E-cadherin in CRC cells in HG-concentration medium (Figure 4E).

### 3.5. Transferring CRC Cell Lines from HG-Concentration Medium to NG-Concentration Medium Affects Cell Proliferation and Morphology

HG concentrations have been shown to be a powerful factor promoting CRC progression. According to the present data, the HG concentration contributed to enhancing the cell proliferation ability and cell cycle progression in SW480 and SW620 cells (Figure 1). Therefore, we examined whether transferring CRC cells from the HG-concentration medium to the NG-concentration medium affected cell proliferation. First, SW480 and SW620 cells were cultured for 10 generations in original HG-concentration medium then transferred to the NG-concentration and medium cultured for 10 generations for comparison. We found that this significantly decreased cell proliferation in SW480 cells by 0.76-fold (*p* < 0.05) and in SW620 cells by 0.38-fold (*p* < 0.05) (Figure 5A,B). Next, we attempted to identify whether the results of transferring CRC cell lines from the HG-concentration medium to the NG-concentration medium were reversible. We determined that this transfer rescued N-cadherin and increased E-cadherin. Furthermore, this effect increased the expression of cell-cycle-regulated cyclin B1 proteins, as determined by Western blotting, and changed the cell morphology to an epithilial type, as observed under a microscope (Figure 5C–E). Overall, our results demonstrate that the effect of HG concentrations is reversible in SW480 and SW620 CRC cells.

### 3.6. E-cadherin Is a Regulator Targeting miR-9 that Was Negatively Correlated with Carcinoembryonic Antigen CEA in Patients with CRC and Hyperglycemia

Based on the aforementioned results, we further examined the negative role of miR-9 in patients with CRC and hyperglycemia. Fasting blood glucose concentration is one measurement that can be used to conduct this screening. For patients with hyperglycemia or diabetes, having one additional risk factor places them at an increased risk of developing CRC. The American Diabetes Association recommends the following screening guideline for fasting glucose: Patients with a fasting sugar level over 126 mg/dL have a high chance of having hyperglycemia or diabetes. Of the patients in our analysis group, 38.5% had prediabetes (≤126 mg/dL; *n* = 5) and 61.5% had hyperglycemia (>126 mg/dL; *n* = 8) (*p* = 0.0119; Figure 6A). Average miR-9 expression levels were determined in 13 surgical specimens of human CRC tissues and compared with those in adjacent normal tissues by using qRT-PCR. A statistically significant decrease in miR-9 was observed in the hyperglycemia group (*p* = 0.0224) (Figure 6B). In the same tissue, the average protein expression of E-cadherin was also decreased in the hyperglycemia group, as determined through WB (*p* = 0.0402; Figure 6C). However, the average expression of cyclin B1 did not change significantly in the CRC specimens from the hyperglycemia group (*p* = 0.2883; Figure 6D). Next, we investigated whether different blood sugar concentrations affect CEA levels in patients with CRC. We found that the average CEA levels did not change significantly in the hyperglycemia group compared with the nondiabetes group (*p* = 0.0824; Figure 6E). We classified CEA levels of ≤5 ng/mL as lower expression and levels of >5 ng/mL as higher expression. As shown in Figure 6F,G, blood sugar concentrations did not have a significant effect on CEA levels, but the average CEA levels were considerably high in CRC tissues. Moreover, we determined that miR-9 was downregulated in CRC tissues with high levels of CEA (*p* = 0.0077; Figure 6H). In particular, the downregulation of miR-9 was determined to be inversely correlated with CEA levels in patients with CRC.

## 4. Discussion

Glucose is an essential nutrient that provides cellular energy homeostasis. Extensive evidence exists that cancer cells are more sensitive to different concentrations of glucose than are normal cells owing to their higher energy consumption ratios [28,29]. Epidemiological evidence suggests that people with hyperglycemia are at a significantly high risk of developing numerous types of cancer [3]. While sufficient bodies of scientific evidence demonstrate the effects of glucose in normal cells, the rigorous molecular mechanisms of glucose in cancer cells are unclear [30,31,32,33]. However, several reports have indicated varying or conflicting results of experiments evaluating the adverse effect of exposure to HG concentrations. HG concentrations can promote cell migration and invasion through the STAT3-induced matrix metalloproteinase-9 (MMP-9) signaling pathways in CT-26 CRC cells [2]. Saengboonmee et al. indicated that HG concentrations enhance the progression of cholangiocarcinoma cells through STAT3 activation [34]. Moreover, HG concentrations increase the degradation of pSTAT3 in Ishikawa endometrial cancer cells and decrease tumor weights in vivo through Metformin [35].

Another crucial factor is how HG concentrations trigger the gene transcription required for mitochondrial functions in tumors. Aerobic glycolysis is combined with various factors, such as oncogenes, tumor suppressors, a hypoxic microenvironment, mitochondrial DNA (mtDNA) mutations, genetic backgrounds, and post-translational modifications, in numerous cancers [36,37,38,39]. These findings illustrate systemic dysfunctions that lead to abnormal cross-talk between hyperglycemia and cancer in the maintenance of cell homeostasis. Studies have demonstrated that hyperglycemia induces increased cell cycle progression and DNA synthesis in colon cancer cells [40,41]. Our data show that high concentrations of d-glucose but not l-glucose could promote cell proliferation ability in SW480 cells (low metastatic) and SW620 (highly metastatic) (Figure 1A). Using a cell analyzer (FACSCalibur, BD Biosciences), we measured DNA synthesis by incorporating propidium iodide after 24 h of serum starvation. HG concentrations increased the G1 population (Figure 1B). Furthermore, the cell-cycle-regulated proteins CDC42, cyclin B1, cyclin D1, and p16 were significantly increased (Figure 1C). Previous studies have demonstrated that cyclin B1 is a key molecule for G2-M phase transition during the cell cycle in CRC. Cyclin B1 and CDC2 were revealed to cooperate positively to play a role in the progression of breast carcinomas, as determined through immunohistochemical (IHC) staining [42]. Another study indicated that cyclin B1 was expressed in different time-window sections of G1 in malignant cancer cells [43]. In colon cancer, p16 expression is mostly elevated, whereas normal tissues exhibit only little or no p16 protein expression [44]. A recent meta-analysis revealed that p16 protein overexpression is associated with the occurrence of CRC in Caucasians. Furthermore, p16 aberrant expression is associated with the Duke stage and lymph-node metastasis of CRC [45]. A recent study showed that the p16^ink4a^ expression was increased in the kidneys of type 2 diabetic patients [46], which suggests that p16 expression may be increased in HG microenvironment. Our results are in line with these reports that p16 expression is elevated; however, the effect of p16 elevation by HG in CRC cells needs further elucidation. According to the preceding discussion, high concentrations of d-glucose play a critical role in promoting CRC cell proliferation.

The EMT is a multifaceted process that is critical for the acquisition of migration, invasiveness, and pluripotent stem cell-like phenotypes. The EMT has been demonstrated to play a role in tumor progression activities through the use of EMT-associated markers, such as mesenchymal-specific markers (i.e., vimentin and N-cadherin) and epithelial-specific markers (i.e., E-cadherin). However, metastatic growth develops when cancer cells become invasive through an altered phenotype, penetrating into the circulatory system and taking hold in distant organs [47]. Therefore, we investigated whether an HG concentration induced EMT characteristics through inducing a mesenchymal morphology (Figure 2A) and increasing the expression of N-cadherin with concomitant decreases in E-cadherin in CRC cells (Figure 2B). Our data indicate that the HG concentration significantly increased CRC cell migration ability (Figure 2C–E). The IGF system involves various regulatory networks that can control numerous developmental and physiological functions, including growth, mitosis, apoptosis, and differentiation [48,49]. Considerable evidence indicates that OSI-906 is a small molecule inhibitor that inhibits IGF1 from binding to IGF1R through autophosphorylation [50]. OSI-906 has been shown to decrease IGF1R, AKT, and ERK phosphorylation [51]. Moreover, previous research showed that Src expression was increased in approximately 80% of CRC specimens compared with normal colonic epithelial specimens and that colorectal metastases exhibited increased activity compared with primary colon tumors [52]. The Src family kinase inhibitor PP1 effectively blocks TGF-β1-induced cell migration and invasion in established PDAC (Panc-1, Colo 357) and primary NSCLC (Tu459) cell lines [53]. Moreover, our study revealed that the HG concentration promoted the expression of p-IGF1R (pY11135/1136), p-Src (pY418), and p-ERK proteins in CRC cells (Figure 2F). These findings provide evidence that the two powerful inhibitors OSI-906 and PP1 inhibited cell proliferation, migration, and invasion in vitro and directly targeted cyclin B1 and N-cadherin (Figure 3A–J). Taken together, the HG concentration regulated environments through multiple processes; it promoted cell proliferation, migration, and invasion and changed the morphology of human CRC cells through regulating the IGF1R/Src/ERK pathways. This indicates that the transduction pathway of tumors or numerous cellular pathways overlap and are likely dependent on the cell type and cell context. In addition, without adding any external pressure or drug stimulation, we determined that transferring cells from an HG-concentration medium to an NG-concentration medium could rescue the cell proliferation level and change the cell morphology to epithelial type (Figure 5A–D). Moreover, a switch from N-cadherin to E-cadherin expression indicated a reversible mesenchymal-to-epithelial transition (Figure 5E). A notable finding is the stepwise recovery for the progress of CRC cells. Collectively, we demonstrated that low-glucose-concentration environments can be used to fight against cancer in the future. Media with low glucose concentrations have already been implemented to a certain extent because of daily necessity. They have been implemented not just alone but also in combination with low-glucose diets and drugs (i.e., Indometacin or MTOB) for therapy [54,55]. Low-glucose diet therapy for patients with CRC seems highly feasible, and managing blood sugar would enable the accurate selection of patients who may benefit from such treatment. These findings imply a novel therapeutic strategy for patients with CRC who simultaneously have hyperglycemia.

Based on our results in CRC cells, we further demonstrated the pivotal role of miR-9 expression in the interplay between high-glucose stimulation and target proteins. Several studies have provided sufficient evidence that miR-9 serves as a tumor suppressor in CRC [19,20,56]. Moreover, miR-9 has been implicated in insulin secretion and has been proposed to be regulated by HG levels in pancreatic beta-cells (insulinoma) [57,58], and recent works show that miR-9a regulates body growth by controlling sNPFR1/NPYR-mediated modulation of insulin signaling [59]. MiR-9 is considered a crucial mediator in cancer development; it works through various biological mechanisms to regulate various targets. The direct targets of microRNA are molecules at transcriptional level; for example, miR-9 directly downregulates TM4SF1 to suppress cell migration and invasion in CRC [60]. In addition, a previous study demonstrated that expression of exogenous miR-9-5p decreased BRAF protein and mRNA levels, and BRAF is a direct target of miR-9-5p in the tumorigenesis of papillary thyroid cancer (PTC) [61]. RAF is crucial in IGF1R/SHC/RAF/MAPK signaling pathway [24]. Taken together, RAF may be a direct target of miR-9 in CRC. In this study, we focused on the influence of a specific HG concentration on cancer proliferation and metastasis through miR-9 in CRC. Accordingly, we investigated the mechanism through which the HG concentration regulated miR-9 in CRC biology. This study demonstrated that the HG concentration significantly reduced endogenous miR-9 levels in SW480 and SW620 cells (Figure 4A,B). Furthermore, Western blotting was employed to validate the downregulation of p-IGF1R (pY11135/1136), cyclin B1, and N-cadherin as well as the upregulation of E-cadherin expression by pre-miR-9 overexpression in SW480 and SW620 cells. A critical finding is that miR-9, as a tumor-suppressive microRNA, operated through the IGF1R pathway to regulate the targeting of cyclin B1 and N-cadherin and the upregulation of E-cadherin in CRC cells in the HG-concentration medium (Figure 4C–E).

We analyzed the relationship between blood sugar levels and miR-9 levels in patients with CRC for relevant prognoses to prove that our findings are valid not only in vitro but also in human CRC. A previous report showed that E-cadherin is a direct target of miR-9 through the inhibition of the NF-κB1-Snail1 pathway in melanoma [62]. Our data reveal that the average miR-9 and E-cadherin expression levels were significantly decreased in patients with hyperglycemia (Figure 6B,C). Thus, E-cadherin may be a direct target of mir-9 in patients with CRC and hyperglycemia simultaneously. This is because preoperative concentrations of CEA may be included in the standard staging procedures for prognosis assessment, thereby improving survival [63,64]. Our data show that the downregulation of miR-9 was correlated with high CEA levels (>5.0 ng/mL) in patients with both CRC and hyperglycemia (Figure 6H). Thus, targeting low-glucose-mediated miR-9 activity may be helpful for the treatment of CRC. Multiple lines of evidence suggest that a substantial proportion of patients with CRC have hyperglycemia, which contributes to CRC progression. A low-sugar diet may provide valuable information for the diagnosis and even prognosis of patients with CRC.

## 5. Conclusions

This study determined that an HG concentration can reduce the tumor-suppressive role of miR-9 in CRC cells. In addition, the HG concentration can promote cell proliferation, increase G1 populations, modulate EMT protein expression and cell morphology, and promote cell migration and invasion ability in SW480 and SW620 CRC cells. The HG concentration also activated the IGF1R/Src axis and upregulated the expression of the ERK, cyclin B1, and N-cadherin signaling pathways through mediating the downregulation of miR-9 expression. Moreover, this study found that miR-9 repressed CRC cell migration ability by increasing E-cadherin, either through another pathway or directly (Figure 7). These findings indicate that hyperglycemia control may serve as a potential strategy for CRC clinical therapy. Moreover, our results provide new evidence that HG concentrations modulate tumor processes through multiple signaling pathways in CRC.

## Figures and Tables

**Figure 1 cells-08-00326-f001:**
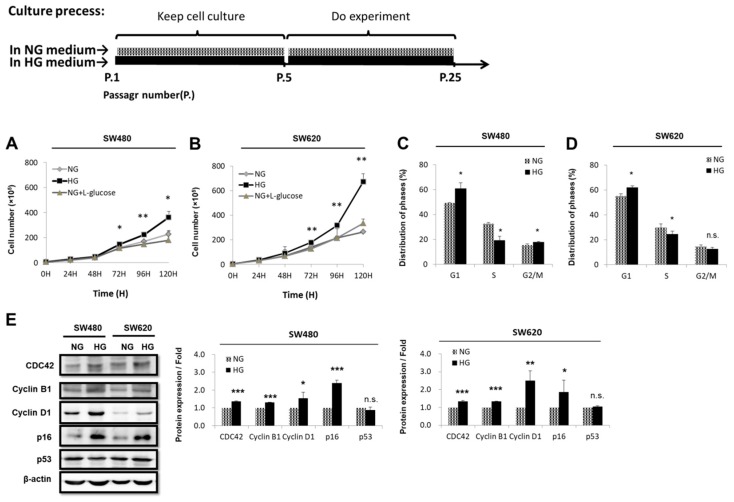
Glucose promoted cell proliferation and induced cell-cycle-regulated protein expression in colorectal cancer (CRC) cells. (**A**) SW480 (low metastatic potential) and (**B**) SW620 (high metastatic potential) cells were cultured in medium with different concentrations of glucose: Normal glucose (NG, 5.5 mM d-glucose), high glucose (HG, 25 mM d-glucose), and osmotic control (NG + l-glucose, 5.5 mM d-glucose + 19.5 mM l-glucose) for a period from 0 to 120 h. Trypan blue stain assay was used to analyze proliferation rates. These data show that d-glucose but not l-glucose promoted cell proliferation. A significant increase in proliferation was observed in CRC cells cultured in HG-concentration medium compared with NG or osmotic control at 72, 96, and 120 h. (**C**,**D**) Cell cycle analysis was performed using FACSCalibur. These data show that HG concentration promoted cell cycle G1 arrest in both cell types. The data are representative of two independent experiments. (**E**) SW480 and SW620 cells were cultured in medium with different concentrations of glucose (NG and HG) for 48 h. The expression levels of CDC42, cyclin B1, cyclin D1, p16, and p53 cell cycle regulated protein were examined using Western blotting. All proteins were increased in HG-concentration medium, but p53 was unchanged in both CRC cell lines. Statistically significant differences between the two groups were judged using Student’s *t*-tests; * *p* < 0.05, ** *p* < 0.005, *** *p* < 0.001; n.s. = nonsignificant.

**Figure 2 cells-08-00326-f002:**
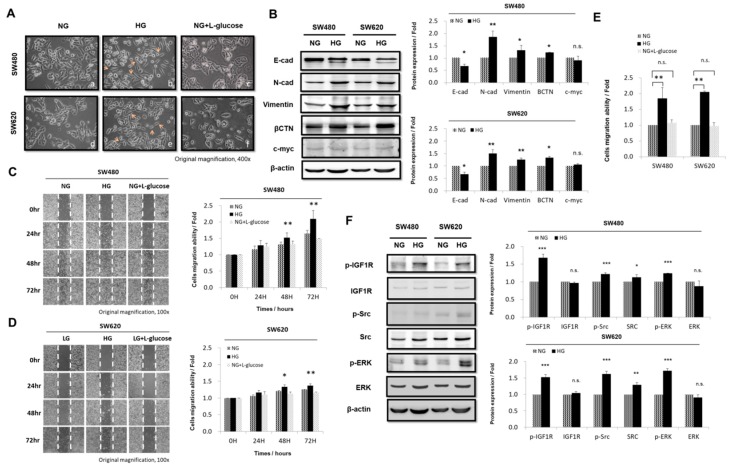
High glucose (HG) concentrations induced epithelial-to-mesenchymal transition protein expression and enhanced migration activity in colorectal cancer (CRC) cells. SW480 (low metastatic potential) and SW620 (high metastatic potential) cells were cultured in different concentrations of glucose (normal: NG; HG; and osmotic control: NG + l-glucose). (**A**) Morphological change occurred from epithelial to mesenchymal type in the HG-concentration group. (**B**) HG concentration caused downregulation of E-cadherin and upregulation of N-cadherin, βCTN, and vimentin, but c-myc was unchanged, as detected using Western blotting. β-actin was evaluated as an internal control. (**C**,**D**) Wound healing assay showed that HG concentration promoted cell motility in SW480 and SW620 CRC cells after 48 and 72 h of culture, compared with the NG and NG + l-glucose groups. (**E**) In a Transwell migration assay, 3.5 × 10^5^ SW480 and SW620 CRC cells were plated onto a 24-well plate and cultured in NG and HG-concentration medium for 96 h. HG concentration promoted cell motility in SW480 and SW620 cells. NG + l-glucose cells were evaluated as ostomic controls. (**F**) These data show that HG concentration caused upregulation of p-IGF1R in CRC. In addition, HG concentration promoted IGF1R downstream signaling, including p-Src and p-ERK; these proteins were increased when CRC cells were cultured in HG-concentration medium. Levels of β-actin were evaluated as loading controls. Statistically significant differences between the two groups were judged using Student’s *t*-tests; * *p* < 0.05, ** *p* < 0.005, *** *p* < 0.001; n.s. = nonsignificant.

**Figure 3 cells-08-00326-f003:**
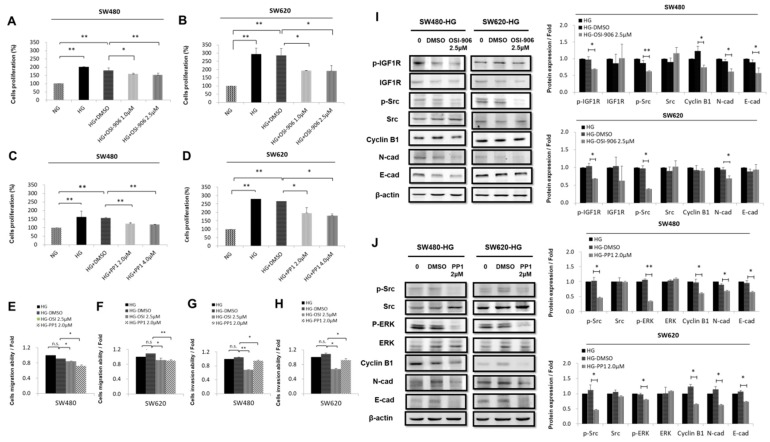
High glucose (HG) concentrations regulated IGF1R and Src and promoted the downstream signaling pathway in colorectal cancer (CRC) cells. (**A**,**B**) OSI-906 (IGF1R inhibitor) or (**C**,**D**) PP1 (Src inhibitor) affected proliferation in a dose-dependent manner in CRC cells. First, 3.5 × 10^5^ SW480 and SW620 cells were seeded onto a 24-well plate. After incubation overnight, they were treated with OSI-906 (1.0 μM and 2.5 μM) or PP1 (2.0 μM and 4.0 μM). These data show that OSI-906 and PP1 significantly inhibited proliferation induced by HG concentration in SW480 and SW620 cells at 1.0 μM and 2.5 μM doses or 2.0 μM and 4.0 μM doses compared with the control group (dimethyl sulfoxide, DMSO). (**E**–**H**) Metastatic activities of CRC cells treated with OSI-906 or PP1 were detected using a Transwell assay; 3.5 × 10^5^ SW480 and SW620 cells were plated onto a 24-well plate and incubated overnight after treatment with OSI-906 2.5 μM or PP1 2.0 μM for 96 h. These data show that OSI-906 (2.5 μM) and PP1 (2 μM) significantly inhibited the migration viability of SW480 and SW620 cells, which was promoted by HG concentration, compared with the HG-concentration group evaluated as positive controls. In addition, 3.5 × 10^5^ SW480 or SW620 cells were cultured in basement membrane matrix-coated 24-well plates with HG-concentration medium and then treated with OSI-906 or PP1 for 168 h. These data show that OSI-906 and PP1 significantly inhibited the invasion viability of SW480 and SW620 cells. (**I**,**J**) Western blot analysis data suggest that OSI-906 or PP1 treatment reduced p-IGF1R or p-Src downstream signaling, reduced the expression of cell-cycle-regulated proteins, and induced and reduced expression of epithelial-to-mesenchymal transition proteins in HG-concentration medium compared with the control group (DMSO). HG concentration was evaluated as a positive control and levels of β-actin were evaluated as loading controls. These data are expressed as mean ± SEM and are representative of two independent experiments according to Student’s *t*-tests; * *p* < 0.05, ** *p* < 0.005; n.s. = nonsignificant.

**Figure 4 cells-08-00326-f004:**
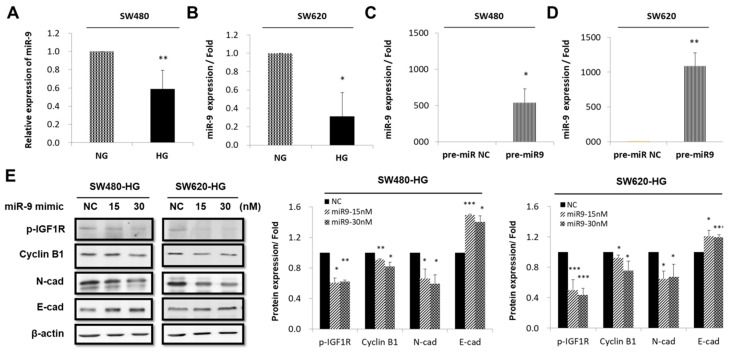
Expression and regulation of miR-9 in colorectal cancer (CRC) cell lines by high glucose (HG) concentration. TaqMan quantitative real-time polymerase chain reaction analysis was performed on miR-9 in SW480 and SW620 CRC cell lines that were cultured in different concentrations of glucose: Namely NG (5.5 mM d-glucose) and HG (25 mM d-glucose). In HG-concentration medium, miR-9 was decreased in both (**A**) SW480 (*p* < 0.005) and (**B**) SW620 (*p* < 0.05). All data were analyzed using a relative quantification method (2^−∆∆Ct^) with RNU6B small RNA as an internal control. (**C**) SW480 and (**D**) SW620 cells were transfected with pre-miR-9 at different doses (15 and 30 nM) or pre-miR negative control (NC) for 48 h in HG-concentration medium. (**E**) Western blotting validated the downregulation of p-IGF1R, cyclin B1, and N-cadherin as well as the upregulation of E-cadherin expression through pre-miR-9 overexpression, compared with pre-miR NC. β-actin was evaluated as an internal control. Statistically significant differences between the two groups were judged using Student’s *t*-tests, * *p* < 0.05, ** *p* < 0.005, *** *p* < 0.001; n.s. = nonsignificant.

**Figure 5 cells-08-00326-f005:**
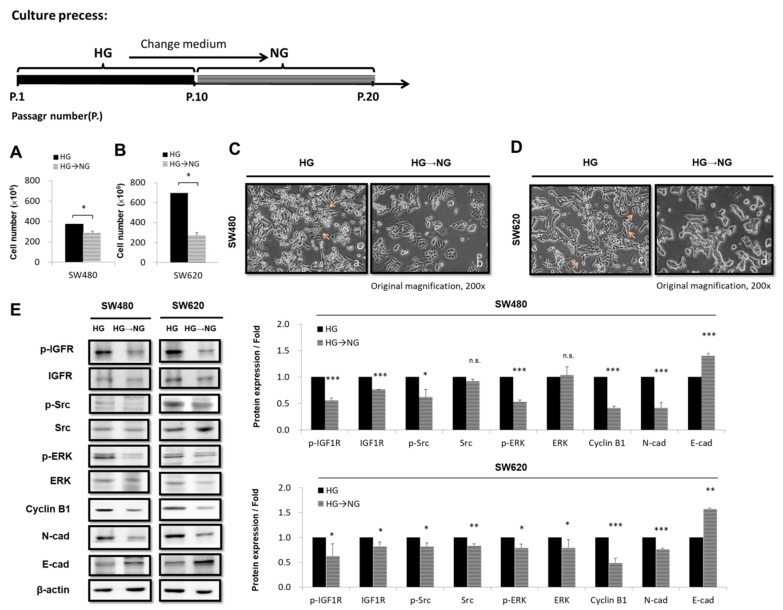
Expression of cell proliferation and morphology was reversible in colorectal cancer (CRC) cell lines by transferring them from high glucose (HG)-concentration medium to normal glucose (NG)-concentration medium. (**A**) SW480 and (**B**) SW620 cells were exposed to medium with different concentrations of glucose, namely NG (5.5 mM d-glucose) and HG (25 mM d-glucose), for a period from 0 to 120 h. Trypan blue stain assay was used to analyze proliferation rates. After 10 generations, a significant rescue of proliferation rate was observed in CRC cells cultured in NG- and HG-concentration media for 120 h compared with the control group. (**C**–**E**) Transferring cells from HG- to NG-concentration medium rescued epithelial-to-mesenchymal transition (EMT) protein marker expression and cell-cycle-regulated protein reversed EMT protein marker expression; cell-cycle-regulated protein reduced cell proliferation and changed cell morphology to an epithelial type. Levels of β-actin were evaluated as loading controls. These data show that the cellular mechanism modulated by glucose concentration in CRC lines is reversible. Statistically significant differences between the two groups were judged by Student’s *t*-tests; * *p* < 0.05, ** *p* < 0.005, *** *p* < 0.001; n.s. = nonsignificant.

**Figure 6 cells-08-00326-f006:**
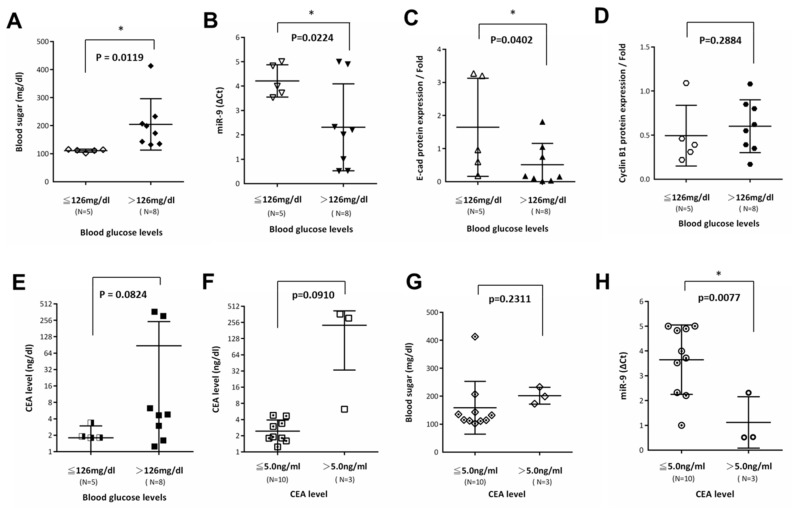
E-cadherin is a regulator targeting miR-9 that was negatively correlated with carcinoembryonic antigen (CEA) in patients with colorectal cancer (CRC) and hyperglycemia. (**A**) Average expression level of blood sugar was upregulated in CRC tissues in the hyperglycemia group (>126 mg/dL, *n* = 8). (**B**) Average expression level of miR-9 was downregulated in CRC specimens in the hyperglycemia group (>126 mg/dL, *n* = 8) as determined by quantitative real-time polymerase chain reaction (qRT-PCR). (**C**) Western blot analysis data show that the average expression of E-cadherin was also decreased in CRC specimens in the hyperglycemia group (>126 mg/dL, *n* = 8). (**D**) However, the average expression of cyclin B1 did not change significantly in CRC specimens in the hyperglycemia group. (**E**) Average CEA levels in the clinical data were increased in the hyperglycemia group (>126 mg/dL, *n* = 8) compared with the prediabetes group (≦125 mg/dL), but the difference did not reach statistical significance. (**F**) Average CEA levels were upregulated in CRC specimens with high CEA expression (CEA > 5.0 ng/mL, *n* = 3) but the difference did not reach statistical significance. (**G**) Average expression level of blood sugar was upregulated in CRC tissues with high CEA expression (CEA > 5.0 ng/mL, *n* = 3), but the difference did not reach statistical significance. (**H**) Average expression level of miR-9 was downregulated in CRC specimens with high CEA expression (CEA > 5.0 ng/mL, *n* = 3) as determined by qRT-PCR. These data show that the effect of miR-9 expression was negatively correlated with the average expression of CEA in CRC specimens. Moreover, E-cadherin was a direct target of miR-9 in CRC. Statistically significant differences between the two groups were judged by Student’s *t*-tests; * *p* < 0.05; n.s. = nonsignificant.

**Figure 7 cells-08-00326-f007:**
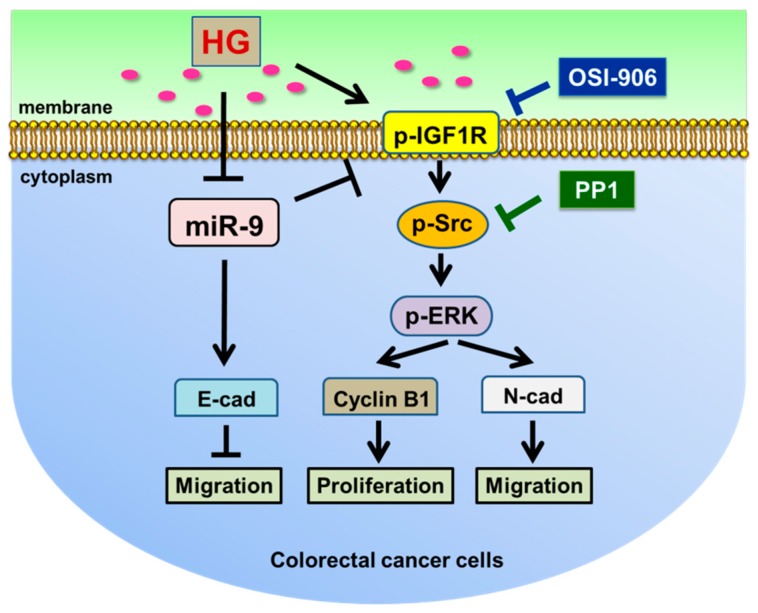
Molecular mechanism through which high glucose (HG) concentration promotes proliferation and migration in colorectal cancer (CRC) cells. HG concentration activated pIGF1R and p-Src expression and increased downstream signaling by mediating the downregulation of miR-9 expression. Moreover, OSI-906 decreased the expression of the EMT protein N-cadherin and reduced the expression of the cell-cycle-regulated protein cyclin B1, as determined through Western blotting, but only cyclin B1 and E-cadherin were unchanged in SW620 cells (Figure 3I). Similarly, PP1 decreased the expression of the EMT protein N-cadherin and reduced the expression of the cell-cycle-regulated protein cyclin B1, as determined through Western blotting (Figure 3J), compared with the control group (dimethyl sulfoxide) cultured in HG-concentration medium. These data demonstrate that HG concentration promoted CRC cell proliferation, modulated EMT protein expression and morphology, and promoted cell migration and invasion ability through the IGF1R/Src/ERK pathway. In addition, miR-9-transfected cells expressed lower levels of p-IGF1R, cyclin B1, and N-cadherin, but E-cadherin was more upregulated compared with the negative control-transfected SW480 and SW620 cells, as determined through Western blotting. Thus, miR-9 is a tumor-suppressive microRNA that may regulate through the IGF1R/Src/ERK pathway the targeting of cyclin B1, N-cadherin, and E-cadherin in CRC cells in an HG-concentration environment (Figure 4E).

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
