# Peer review of "High Glucose Concentrations Negatively Regulate the IGF1R/Src/ERK Axis through the MicroRNA-9 in Colorectal Cancer"

_cells, 2019, doi:10.3390/cells8040326_

Reviewer 1 Report

The authors have done a nice job of convincing me that their story is true and relevant. The amount of work is sufficient and convincing. I believe the manuscript will be acceptable for publication pending manuscript editing for English usage and grammar, etc.

For instance, "Aberrant expression" as indicated in the abstract  would typically refer to high expression to most English readers. This should be rewritten, for instance as "aberrant reduced expression" to clarify that High glucose causes reduced miR-( expression).

Glu should not be used as an abbreviation for glucose as it is a standard three letter English abbreviation for glutamate.

Author Response

Comment 1: [The authors have done a nice job of convincing me that their story is true and relevant. The amount of work is sufficient and convincing. I believe the manuscript will be acceptable for publication pending manuscript editing for English usage and grammar, etc. For instance, "Aberrant expression" as indicated in the abstract would typically refer to high expression to most English readers. This should be rewritten, for instance as "aberrant reduced expression" to clarify that High glucose causes reduced miR-( expression). ]

Response 1: Thank you for your comments. We have incorporated your suggestion throughout the manuscript.

Comment 2: [Glu should not be used as an abbreviation for glucose as it is a standard three letter English abbreviation for glutamate. ]

Response 2: Thank you for your valuable comment. We have modified this word.

Reviewer 2 Report

In this manuscript, high glucose was found to attenuate miR-9, in turn reduced IGF1R signaling and cell migration and proliferation in colorectal cancer cells. To this reviewer, the justification for focus only on miR-9 not other miRs is relatively weak. Are there any previous studies on the regulation of miR-9 on IGF1R or Src? If so, the authors need to cite the references. If not, the authors need to provide justification why targeting miR-9/IGF1R, but not other candidates.  

In Fig 1C and D, one can argue that HG induces G1 arrest rather than enhancement of cell proliferation. It is suggested to analyze cell doubling time, for that can be better explain whether HG indeed stimulates cell growth. 

Can the authors discuss more in details why p16 expression was significantly high in HG treatment in both cell lines? 

It seems like there is no significantly difference in cell migration in HG groups compared to the control at 24h. Since the authors claimed that HG treatment also enhance cell proliferation, it is extremely important to validate that the enhanced cell migration at 48 and 72 h is not due to higher cell numbers.   

If HG-mediated miR-9 is responsible for the regulation of IGF1R, should we observe more marked up-regulation of IGF1R protein in Figure 2F? It is not clear why the miR expression would affect protein phosphorylation status rather than the protein expression itself.   

 Other protein targets of miR-9 should be included in this study to validate that IGF1R is indeed involved in HG-mediated cell migration/proliferation. 

It is suggested to use more updated references, such as the papers on the relationship of hyperglycemia and CRC, etc. 

Author Response

Comment 1: [In this manuscript, high glucose was found to attenuate miR-9, in turn reduced IGF1R signaling and cell migration and proliferation in colorectal cancer cells. To this reviewer, the justification for focus only on miR-9 not other miRs is relatively weak. Are there any previous studies on the regulation of miR-9 on IGF1R or Src? If so, the authors need to cite the references. If not, the authors need to provide justification why targeting miR-9/IGF1R, but not other candidates. ]

Response 1: Thank you for pointing this out. So far, we were rejoiced, i am sure no any previous studies on the regulation of miR-9 on IGF1R or Src. Several studies have provided sufficient evidence that miR-9 serves as a tumor suppressor in CRC [19,20,53]. In additional to, miR-9 has been implicated in insulin secretion and has been proposed to be regulated by HG levels in pancreatic beta-cells (insulinoma) [54,55]. Based on our results in CRC cells, we further clarified the pivotal role of miR-9 expression in the interplay between high-glucose stimulation and target proteins. Moreover, hyperglycemia may increase circulating insulin-like growth factor-1 (IGF-1) levels through ligands binding to insulin receptors [21,22]. In parallel, IGF-1 receptor (IGF1R) is an autophosphorylated receptor that binds to the Src homology domain and insulin receptor substrate (IRS). In this study, we focused on the influence of a specific HG concentration on cancer proliferation and metastasis through miR-9-IGF1R or Src pathway in CRC.

References:

19.    Cekaite, L.; Rantala, J.K.; Bruun, J.; Guriby, M.; Agesen, T.H.; Danielsen, S.A.; Lind, G.E.; Nesbakken, A.; Kallioniemi, O.; Lothe, R.A., et al. Mir-9, -31, and -182 deregulation promote proliferation and tumor cell survival in colon cancer. Neoplasia 2012, 14, 868-879.

20.    Bandres, E.; Agirre, X.; Bitarte, N.; Ramirez, N.; Zarate, R.; Roman-Gomez, J.; Prosper, F.; Garcia-Foncillas, J. Epigenetic regulation of microrna expression in colorectal cancer. Int. J. Cancer 2009, 125, 2737-2743. 10.1002/ijc.24638

21.    Roberts, D.L.; Dive, C.; Renehan, A.G. Biological mechanisms linking obesity and cancer risk: New perspectives. Annu. Rev. Med. 2010, 61, 301-316. 10.1146/annurev.med.080708.082713

22.    Handelsman, Y.; Leroith, D.; Bloomgarden, Z.T.; Dagogo-Jack, S.; Einhorn, D.; Garber, A.J.; Grunberger, G.; Harrell, R.M.; Gagel, R.F.; Lebovitz, H.E., et al. Diabetes and cancer--an aace/ace onsensus statement. Endocr. Pract. 2013, 19, 675-693. 10.4158/ep13248.cs

53.    Oberg, A.L.; French, A.J.; Sarver, A.L.; Subramanian, S.; Morlan, B.W.; Riska, S.M.; Borralho, P.M.; Cunningham, J.M.; Boardman, L.A.; Wang, L., et al. Mirna expression in colon polyps provides evidence for a multihit model of colon cancer. PLoS One 2011, 6, e20465. 10.1371/journal.pone.0020465

54.    Ramachandran, D.; Roy, U.; Garg, S.; Ghosh, S.; Pathak, S.; Kolthur-Seetharam, U. Sirt1 and mir-9 expression is regulated during glucose-stimulated insulin secretion in pancreatic beta-islets. Febs j 2011, 278, 1167-1174. 10.1111/j.1742-4658.2011.08042.x

55.    Hu, D.; Wang, Y.; Zhang, H.; Kong, D. Identification of mir-9 as a negative factor of insulin secretion from beta cells. J. Physiol. Biochem. 2018, 74, 291-299. 10.1007/s13105-018-0615-3

Comment 2: [In Fig 1C and D, one can argue that HG induces G1 arrest rather than enhancement of cell proliferation. It is suggested to analyze cell doubling time, for that can be better explain whether HG indeed stimulates cell growth. ]

Response 2: Thank you for this comment. It would have been interesting to explore this aspect. Previous studies have demonstrated that HG-induced cell cycle arrest is considered to be the main cause of the development of diabetic complicationsPMID:23642823. However, cyclin B1 is a key molecule for G2-M phase transition during the cell cycle in CRC. Cyclin B1 and CDC2 were revealed to cooperate positively to play a role in the progression of breast carcinomas, as determined through immunohistochemical (IHC) staining [42]. Another study indicated that cyclin B1 was expressed in different time-window sections of G1 in malignant cancer cells [43]. Our data clearly demonstrate the HG concentration increased cell proliferation through enhanced G1 population progression in Figure 1C and 1D.

References:

(PMID:23642823)

Yu, X.Y.; Geng, Y.J.; Lei, H.P.; Lin, Q.X.; Yuan, J.; Li, Y. Igf-1 prevents high glucose-induced cell cycle arrest in cardiomyocytes via beta-catenin pathway. Int. J. Cardiol. 2013, 168, 2869-2870. 10.1016/j.ijcard.2013.03.145

42. Chae, S.W.; Sohn, J.H.; Kim, D.H.; Choi, Y.J.; Park, Y.L.; Kim, K.; Cho, Y.H.; Pyo, J.S.; Kim, J.H. Overexpressions of cyclin b1, cdc2, p16 and p53 in human breast cancer: The clinicopathologic correlations and prognostic implications. Yonsei Med. J. 2011, 52, 445-453. 10.3349/ymj.2011.52.3.445

43. Shen, M.; Feng, Y.; Gao, C.; Tao, D.; Hu, J.; Reed, E.; Li, Q.Q.; Gong, J. Detection of cyclin b1 expression in g(1)-phase cancer cell lines and cancer tissues by postsorting western blot analysis. Cancer Res. 2004, 64, 1607-1610. 10.1158/0008-5472

Comment 3: [Can the authors discuss more in details why p16 expression was significantly high in HG treatment in both cell lines? ]

Response 3: Thank you for your comment. It would have been interesting to explore this aspect. In colon cancer, p16 expression is mostly elevated, whereas normal tissues exhibit only little or no protein expression (PMID: 11040180). Previous studies concentrated on the inactivating mechanisms of the tumor suppressor gene, such as methylation of promoter or 5' regulator regions. In some cases, methylation of p16 correlated with shorter survival or worse prognosis (PMID: 30348132). No matter what reason may lead to the unscheduled expression of p16. The presence of p16 may suggest a possible dysfunction or misregulation of the machinery for cell cycle progression in HG treatment in CRC cells.

References:

(PMID: 11040180)

Dai, C.Y.; Furth, E.E.; Mick, R.; Koh, J.; Takayama, T.; Niitsu, Y.; Enders, G.H. P16(ink4a) expression begins early in human colon neoplasia and correlates inversely with markers of cell proliferation. Gastroenterology 2000, 119, 929-942.

(PMID: 30348132)

Ye, X.; Mo, M.; Xu, S.; Yang, Q.; Wu, M.; Zhang, J.; Chen, B.; Li, J.; Zhong, Y.; Huang, Q., et al. The hypermethylation of p16 gene exon 1 and exon 2: Potential biomarkers for colorectal cancer and are associated with cancer pathological staging. BMC Cancer 2018, 18, 1023. 10.1186/s12885-018-4921-5

Comment 4: [It seems like there is no significantly difference in cell migration in HG groups compared to the control at 24h. Since the authors claimed that HG treatment also enhance cell proliferation, it is extremely important to validate that the enhanced cell migration at 48 and 72 h is not due to higher cell numbers. ]

Response 4: Thank you for the valuable comment. We can check proliferation rate at 24 and 48 h was not significantly statistically difference under the HG conditions (Figure 1A and 1B). But, HG concentration significantly promoted SW480 and SW620 cell motility compared with the NG and NG+L-glucose groups after 48 and 72 h of culture by a wound healing assay (Figure 2C and D). These data clearly demonstrate that HG concentration enhanced migration activity in CRC cells.

Comment 5: [If HG-mediated miR-9 is responsible for the regulation of IGF1R, should we observe more marked up-regulation of IGF1R protein in Figure 2F? It is not clear why the miR expression would affect protein phosphorylation status rather than the protein expression itself. ]

Response 5: Thank you for pointing this out. Previous study indicated that miR-9 has been implicated in insulin secretion and has been proposed to be regulated by HG levels in pancreatic beta-cells (insulinoma) [54,55]. IGF-1 receptor (IGF1R) is an autophosphorylated receptor that binds to the Src homology domain and insulin receptor substrate (IRS). IGF1R activates the IRS protein SHC, which then stimulates Raf before triggering a kinase cascade, eventually resulting in the activation of mitogen-activated protein kinases(MAPKs) and extracellular signal-regulated kinases 1 and 2 (ERK1 and ERK2, respectively) through the GTPase Ras [24]. Likewise, receptors with the intracellular domain of IRS display higher IRS phosphorylation, stronger regulation of genes in metabolic pathways and more dramatic glycolytic responses to hormonal stimulation (PMID:28345670). And i am sure no any previous studies on the regulation of miR-9 on IGF1R. According to our speculation, miR-9 expression might directly affect IGF1R protein phosphorylation status rather than the protein expression itself by HG concentration (Figure 4E).

References:

24.    Zha, J.; Lackner, M.R. Targeting the insulin-like growth factor receptor-1r pathway for cancer therapy. Clin. Cancer Res. 2010, 16, 2512-2517. 10.1158/1078-0432.ccr-09-2232.

54.    Ramachandran, D.; Roy, U.; Garg, S.; Ghosh, S.; Pathak, S.; Kolthur-Seetharam, U. Sirt1 and mir-9 expression is regulated during glucose-stimulated insulin secretion in pancreatic beta-islets. Febs j 2011, 278, 1167-1174. 10.1111/j.1742-4658.2011.08042.x

55.    Hu, D.; Wang, Y.; Zhang, H.; Kong, D. Identification of mir-9 as a negative factor of insulin secretion from beta cells. J. Physiol. Biochem. 2018, 74, 291-299. 10.1007/s13105-018-0615-3

(PMID:28345670)

O'Neill, B.T.; Bhardwaj, G.; Penniman, C.M.; Krumpoch, M.T.; Suarez Beltran, P.A.; Klaus, K.; Poro, K.; Li, M.; Pan, H.; Dreyfuss, J.M., et al. Foxo transcription factors are critical regulators of diabetes-related muscle atrophy. Diabetes 2019, 68, 556-570. 10.2337/db18-0416

Comment 6: [It is suggested to use more updated references, such as the papers on the relationship of hyperglycemia and CRC, etc. ]

Response 6: Thank you for the comment. Therefore, we have updated part of the references, as follows:

3.       Lee, M.Y.; Lin, K.D.; Hsiao, P.J.; Shin, S.J. Thiamine corrects delayed replication and decreases production of lactate and advanced glycation end-products in bovine retinal and human umbilical vein endothelial cells cultured under high glucose conditions. Metabolism 2012, 61, 242-249. 10.1016/j.metabol.2011.06.020

4.       Zhu, B.; Wu, X.; Wu, B.; Pei, D.; Zhang, L.; Wei, L. The relationship between diabetes and colorectal cancer prognosis: A meta-analysis based on the cohort studies. PLoS One 2017, 12, e0176068-e0176068. 10.1371/journal.pone.0176068

6.       Vulcan, A.; Manjer, J.; Ohlsson, B. High blood glucose levels are associated with higher risk of colon cancer in men: A cohort study. BMC Cancer 2017, 17, 842. 10.1186/s12885-017-3874-4

8.       Hou, Y.; Zhou, M.; Xie, J.; Chao, P.; Feng, Q.; Wu, J. High glucose levels promote the proliferation of breast cancer cells through gtpases. Breast cancer (Dove Medical Press) 2017, 9, 429-436. 10.2147/bctt.s135665

9.       Ding, C.Z.; Guo, X.F.; Wang, G.L.; Wang, H.T.; Xu, G.H.; Liu, Y.Y.; Wu, Z.J.; Chen, Y.H.; Wang, J.; Wang, W.G. High glucose contributes to the proliferation and migration of non-small cell lung cancer cells via gas5-trib3 axis. Biosci. Rep. 2018, 38, BSR20171014. 10.1042/bsr20171014

11.    Tan, W.; Liu, B.; Qu, S.; Liang, G.; Luo, W.; Gong, C. Micrornas and cancer: Key paradigms in molecular therapy. Oncol. Lett. 2018, 15, 2735-2742. 10.3892/ol.2017.76383030.

30.    Sikander, M.; Malik, S.; Chauhan, N.; Khan, P.; Kumari, S.; Kashyap, V.K.; Khan, S.; Ganju, A.; Halaweish, F.T.; Yallapu, M.M., et al. Cucurbitacin d reprograms glucose metabolic network in prostate cancer. Cancers (Basel) 2019, 11. 10.3390/cancers11030364

31.    Shen, J.; Liu, M.; Xu, J.; Sun, B.; Xu, H.; Zhang, W. Arl15 overexpression attenuates high glucose-induced impairment of insulin signaling and oxidative stress in human umbilical vein endothelial cells. Life Sci. 2019, 220, 127-135. 10.1016/j.lfs.2019.01.030

32.    Chen, S.; Ma, J.; Zhu, H.; Deng, S.; Gu, M.; Qu, S. Hydroxysafflor yellow a attenuates high glucose-induced human umbilical vein endothelial cell dysfunction. Hum. Exp. Toxicol. 2019, 960327119831065. 10.1177/0960327119831065

39.    Petrova, V.; Annicchiarico-Petruzzelli, M.; Melino, G.; Amelio, I. The hypoxic tumourmicroenvironment. Oncogenesis 2018, 7, 10-10. 10.1038/s41389-017-0011-9

44.    Dongre, A.; Weinberg, R.A. New insights into the mechanisms of epithelial–mesenchymal transition and implications for cancer. Nature Reviews Molecular Cell Biology 2019, 20, 69-84. 10.1038/s41580-018-0080-4

 55. Hu, D.; Wang, Y.; Zhang, H.; Kong, D. Identification of mir-9 as a negative factor of insulin secretion from beta cells. J. Physiol. Biochem. 2018, 74, 291-299. 10.1007/s13105-018-0615-3

Round  2

Reviewer 2 Report

It is clear that HG concentrations modulate several cell functions associating with EMT and certain signaling pathways. However, all the phenomena observed seem like taking longer than 48 hours to be significant. The authors have correlated the downregulation of p-IGF1R by the action of miR-9 as main target(s) in this system, to this reviewer, those effects seem to be indirect. Can the authors propose/provide other protein target(s) that might be more directly regulated by miR-9, especially at transcriptional level? 

Could other downstream targets of miR-9 reported in the literature be more directly associated with the effects of HG? or other protein targets in the IGF1R/Src/ERK axis be more directly linked to the effects of HG? 

In this revision, the authors provide citations suggesting that p16 is hypermethylated contributing to cancer. However, if I am not mistaken, the hypermethylated p16 would result in a low protein expression that definitely is not the case observed in Figure 1E.  

Author Response

Reviewer’s comment

Comment 1: [It is clear that HG concentrations modulate several cell functions associating with EMT and certain signaling pathways. However, all the phenomena observed seem like taking longer than 48 hours to be significant. The authors have correlated the downregulation of p-IGF1R by the action of miR-9 as main target(s) in this system, to this reviewer, those effects seem to be indirect. Can the authors propose/provide other protein target(s) that might be more directly regulated by miR-9, especially at transcriptional level?  ]

Response 1: Thank you for pointing this out. Previous studies show that miR-9 in mammals is expressed in both the brain and the pancreatic beta cells where it has been shown to regulate glucose levels via its targets onecut2 and sirt1 (PMID: 16831872, PMID: 21288303). Moreover, neuropeptides also play important roles in the physiology and behaviours of animals. In Drosophila, short neuropeptide F (sNPF) regulates body growth by binding to its receptor sNPFR1 in the insulin-producing cells (IPCs) and activating ERK-mediated Dilp expression (PMID: 18344986, PMID: 15385546). NPY is also released by the autonomic nervous system into the pancreas where it directly modulates insulin and glucagon secretion (PMID: 3914635). And recent works show that both fly sNPFR1 and mammalian NPY2R are targets of miR-9a/miR-9 by demonstrating direct binding of miR-9a/miR-9 to the sNPFR1 and NPY2R 3'-UTRs using an in vitro binding assay and a CLIP assay. This finding indicate that miR-9a regulates body growth by controlling sNPFR1/NPYR-mediated modulation of insulin signaling (PMID: 26138755).

References:

(PMID: 16831872)

Plaisance, V.; Abderrahmani, A.; Perret-Menoud, V.; Jacquemin, P.; Lemaigre, F.; Regazzi, R. Microrna-9 controls the expression of granuphilin/slp4 and the secretory response of insulin-producing cells. J. Biol. Chem. 2006, 281, 26932-26942. 10.1074/jbc.M601225200

(PMID: 21288303)

Ramachandran, D.; Roy, U.; Garg, S.; Ghosh, S.; Pathak, S.; Kolthur-Seetharam, U. Sirt1 and mir-9 expression is regulated during glucose-stimulated insulin secretion in pancreatic beta-islets. The FEBS journal 2011, 278, 1167-1174. 10.1111/j.1742-4658.2011.08042.x

(PMID: 18344986)

Lee, K.S.; Kwon, O.Y.; Lee, J.H.; Kwon, K.; Min, K.J.; Jung, S.A.; Kim, A.K.; You, K.H.; Tatar, M.; Yu, K. Drosophila short neuropeptide f signalling regulates growth by erk-mediated insulin signalling. Nat. Cell Biol. 2008, 10, 468-475. 10.1038/ncb1710

(PMID: 15385546)

Lee, K.S.; You, K.H.; Choo, J.K.; Han, Y.M.; Yu, K. Drosophila short neuropeptide f regulates food intake and body size. J. Biol. Chem. 2004, 279, 50781-50789. 10.1074/jbc.M407842200

(PMID: 3914635)

Moltz, J.H.; McDonald, J.K. Neuropeptide y: Direct and indirect action on insulin secretion in the rat. Peptides 1985, 6, 1155-1159.

(PMID: 26138755)

Suh, Y.S.; Bhat, S.; Hong, S.H.; Shin, M.; Bahk, S.; Cho, K.S.; Kim, S.W.; Lee, K.S.; Kim, Y.J.; Jones, W.D., et al. Genome-wide microrna screening reveals that the evolutionary conserved mir-9a regulates body growth by targeting snpfr1/npyr. Nature communications 2015, 6, 7693. 10.1038/ncomms8693

Comment 2: [Could other downstream targets of miR-9 reported in the literature be more directly associated with the effects of HG? or other protein targets in the IGF1R/Src/ERK axis be more directly linked to the effects of HG? ]

Response 2: Thank you for your valuable comments. In recent years, some mechanisms have been hypothesized to explore how an excess level of glucose affected the occurrence and development of tumors, such as breast, colorectal, pancreatic, and lung cancer [PMID: 28470916, PMID: 27535548, PMID: 26197187, PMID: 29367413]. HG can promote the proliferation, migration, and invasion of cancer cells through various mechanisms. Hyperglycemia, through ligands binding to insulin receptors, may increase circulating insulin-like growth factor-1 (IGF-1) levels [21,22]. IGF-1 was implicated as a key factor in the mechanisms involved in carcinogenesis [23]. In parallel, IGF-1 receptor (IGF1R) is an autophosphorylated receptor that binds to the Src homology domain and insulin receptor substrate (IRS). IGF1R activates the IRS protein SHC, which then stimulates RAF before triggering a kinase cascade, eventually resulting in the activation of mitogen-activated protein kinases(MAPKs) and extracellular signal-regulated kinases 1 and 2 (ERK1 and ERK2, respectively) through the GTPase Ras [24]. Src overexpression has been shown to increase cell adhesion, invasion, and migration in CRC cells. In addition, ERK1/2 may influence transcriptional factors, leading to increased cell cycle activity and promoting cancer progression [25,26]. It has been demonstrated that mutations of the BRAF gene induce uncontrolled and persistent activation of kinase signaling pathways, causing over-proliferation, and differentiation into cancer cells (PMID: 26878440). Previous study demonstrated that expression of exogenous miR-9-5p decreased BRAF protein and mRNA levels, while suppression of endogenous miR-9-5p resulted in an increase in BRAF protein, and mRNA levels. Therefore, BRAF is a direct target of miR-9-5p in the tumorigenesis of papillary thyroid cancer (PTC) (PMID: 30333891). In our study, we focused on whether a specific HG concentration can influence cancer cell proliferation and metastasis in CRC through the IGF1R or Src pathway″ here. Thus, miR-9 is a tumor-suppressive microRNA that may regulate through the IGF1R/Src/ERK pathway the targeting of cyclin B1, N-cadherin, and E-cadherin in CRC cells in an HG-concentration environment (Figure 4E).

References:

(PMID: 28470916)

Obata A.; Okauchi S.; Kimura T.; Hirukawa H.; Tanabe A.; Kinoshita T. Advanced breast cancer in a relatively young man with severe obesity and type 2 diabetes mellitus. J. Diabetes Investig. 2017, 8, 395–396. 10.1111/jdi.12570

(PMID: 27535548)

Yu F.; Guo Y.; Wang H.Feng J.; Jin Z.; Chen Q. Type 2 diabetes mellitus and risk of colorectal adenoma: a meta-analysis of observational studies. BMC Cancer. 2016, 16, 642. 10.1186/s12885-016-2685-3

(PMID: 26197187)

Lewis J.D.; Habel L.A.; Quesenberry C.P.; Strom B.L.; Peng T.; Hedderson M.M. Pioglitazone use and risk of bladder cancer and other common cancers in persons with diabetes. JAMA. 2015, 314, 265–277. 10.1001/jama.2015.7996

(PMID: 29367413)

Ding, C.Z.; Guo, X.F.; Wang, G.L.; Wang, H.T.; Xu, G.H.; Liu, Y.Y.; Wu, Z.J.; Chen, Y.H.; Wang, J.; Wang, W.G. High glucose contributes to the proliferation and migration of non-small cell lung cancer cells via gas5-trib3 axis. Biosci. Rep. 2018. 10.1042/bsr20171014

(PMID: 26878440)

Kundu, A.; Quirit, J.G.; Khouri, M.G.; Firestone, G.L. Inhibition of oncogenic braf activity by indole-3-carbinol disrupts microphthalmia-associated transcription factor expression and arrests melanoma cell proliferation. Mol. Carcinog. 2017, 56, 49-61. 10.1002/mc.22472

(PMID: 30333891)

Guo, F.; Hou, X.; Sun, Q. Microrna-9-5p functions as a tumor suppressor in papillary thyroid cancer via targeting braf. Oncol. Lett. 2018, 16, 6815-6821. 10.3892/ol.2018.9423

21.   Roberts, D.L.; Dive, C.; Renehan, A.G. Biological mechanisms linking obesity and cancer risk: New perspectives. Annu. Rev. Med. 2010, 61, 301-316. 10.1146/annurev.med.080708.082713

22.   Handelsman, Y.; Leroith, D.; Bloomgarden, Z.T.; Dagogo-Jack, S.; Einhorn, D.; Garber, A.J.; Grunberger, G.; Harrell, R.M.; Gagel, R.F.; Lebovitz, H.E., et al. Diabetes and cancer--an aace/ace consensus statement. Endocr. Pract. 2013, 19, 675-693. 10.4158/ep13248.cs

23.   Gristina, V.; Cupri, M.G.; Torchio, M.; Mezzogori, C.; Cacciabue, L.; Danova, M. Diabetes and cancer: A critical appraisal of the pathogenetic and therapeutic links. Biomed Rep 2015, 3, 131-136. 10.3892/br.2014.399

24.   Zha, J.; Lackner, M.R. Targeting the insulin-like growth factor receptor-1r pathway for cancer therapy. Clin. Cancer Res. 2010, 16, 2512-2517. 10.1158/1078-0432.ccr-09-2232

25.   Lieu, C.; Kopetz, S. The src family of protein tyrosine kinases: A new and promising target for colorectal cancer therapy. Clin. Colorectal Cancer 2010, 9, 89-94. 10.3816/CCC.2010.n.012

26.   Chen, J.; Elfiky, A.; Han, M.; Chen, C.; Saif, M.W. The role of src in colon cancer and its therapeutic implications. Clin. Colorectal Cancer. 2014, 13, 5-13. 10.1016/j.clcc.2013.10.003

Comment 3: [In this revision, the authors provide citations suggesting that p16 is hypermethylated contributing to cancer. However, if I am not mistaken, the hypermethylated p16 would result in a low protein expression that definitely is not the case observed in Figure 1E. ]

Response 3 Thank you for your comment. It would be interesting to explore this aspect. Accounting for the given suggestions, we have now revised the description and application of about the p16 expression. In colon cancer, p16 expression is mostly elevated, whereas normal tissues exhibit only little or no p16 protein expression (PMID: 11040180). A recent meta-analysis revealed that p16 protein overexpression is associated with the occurrence of CRC in Caucasians. Furthermore, p16 aberrant expression is associated with the Duke stage and lymph-node metastasis of CRC (PMID: 29561443). A recent study showed that the p16ink4a expression was increased in the kidneys of type 2 diabetic patients (PMID: 26022507), which suggests that p16 expression may be increases in HG microenvironment. Our results are in line with these reports that p16 expression is elevated; however, the effect of p16 elevation by HG in CRC cells needs further elucidation.

References:

PMID: 11040180

Dai, C.Y.; Furth, E.E.; Mick, R.; Koh, J.; Takayama, T.; Niitsu, Y.; Enders, G.H. P16(ink4a) expression begins early in human colon neoplasia and correlates inversely with markers of cell proliferation. Gastroenterology 2000, 119, 929-942.

PMID: 29561443

Zhou, N.; Gu, Q. Prognostic and clinicopathological value of p16 protein aberrant expression in colorectal cancer: A prisma-compliant meta-analysis. Medicine (Baltimore) 2018, 97, e0195. 10.1097/md.0000000000010195

PMID: 26022507

Zhang, Y.Y.; Guo, Q.Y.; Wu, M.Y.; Zang, C.S.; Ma, F.Z.; Sun, T.; Wang, W.N.; Miao, L.N.; Xu, Z.G. P16ink4a expression is increased through 12-lipoxygenase in high glucose-stimulated glomerular mesangial cells and type 2 diabetic glomeruli. Nephron 2015, 130, 141-150. 10.1159/000431106